# Therapeutic Applications of Adeno-Associated Virus (AAV) Gene Transfer of HLA-G in the Eye

**DOI:** 10.3390/ijms23073465

**Published:** 2022-03-23

**Authors:** Brian C. Gilger, Matthew L. Hirsch

**Affiliations:** 1Department of Clinical Sciences, North Carolina State University, Raleigh, NC 27607, USA; 2Ophthalmology, School of Medicine, University of North Carolina, Chapel Hill, NC 27599, USA; mhirsch@email.unc.edu

**Keywords:** AAV, HLA-G, dry eye, ocular graft vs. host disease, corneal transplant rejection, uveitis, gene therapy

## Abstract

The purpose of this paper is to review human leukocyte antigen G (HLA-G) in the eye, its role in immune tolerance, and the potential therapeutic use of AAV gene transfer and expression of HLA-G in various ocular tissues. Several studies are reviewed that demonstrate efficacy in animal models of disease, including intracorneal delivery of AAV-HLA-G to treat corneal inflammation and prevent corneal graft rejection, subconjunctival injection of AAV-HLA-G for ocular graft vs. host disease and potentially dry eye disease, and intravitreal injection of AAV-HLA-G to inhibit uveitis. Furthermore, due to the anti-vascular function of HLA-G, AAV-HLA-G may be an effective therapy for posterior ocular diseases, such as neovascular age-related macular degeneration, diabetic retinopathy, and choroidal neovascularization. Therefore, AAV-mediated gene transfer of HLA-G may be an effective treatment for common immune-mediated, inflammatory, and neovascular diseases of the eye.

## 1. Introduction

The eye is considered an immune privileged organ due, in part, to the blood–ocular barrier that limits blood products, including neutrophils and lymphocytes, from entering the eye. In addition, the eye has an active immune-suppressive mechanism, called the anterior chamber-associated immune deviation (ACAID), which suppresses the adaptive immune response to antigens presented in the eye [1]. Various substances also contribute to this immune-suppressive microenvironment in the eye, including interleukin 10 (IL-10), TGF-β, αMSH, and human leukocyte antigen G (HLA-G), all of which reduce inflammation and induce monocytes and lymphocytes to be immunotolerant [1,2,3]. These physical, chemical, and immunotolerant activities work together to suppress inflammation and vascularization that could be detrimental to vision.

When ocular immune tolerance breaks down, generally with ocular injury, infection, or as a consequence of genetic disease, innate and adaptive immune responses can develop, resulting in a disruption of the blood–ocular barrier and an influx of immune and inflammatory cells into the eye. These upregulated immunologic and antigen-presenting cells work to counteract the insults, and in doing so, may react to self-antigens to develop immune-mediated disease(s) in some individuals. The resulting ocular diseases are typically chronic or relapsing and include dry eye disease (DED), non-infectious uveitis (NIU), and neovascular age-related macular degeneration [4]. These diseases are common, cause blindness, and are difficult to treat. Conventional treatments include ocular or systemic corticosteroids or calcineurin inhibitors, which are only partially effective and have side effects [5,6]. These side effects, poor therapeutic responses, and inconsistent patient treatment compliance all suggest that more-effective and/or sustained treatments are needed for these chronic and blinding inflammation-related ocular diseases.

One solution for treatment of these chronic ocular diseases is gene transfer using an adeno-associated virus (AAV) vector. AAV gene transfer has the advantage of efficient gene delivery and long-term transgene production following a single dose. The AAV protein capsid packages a single-strand DNA molecule and is among the smallest viruses identified to date. AAV is generally considered non-pathogenic, and naturally occurring serotypes have been isolated from most animal species. Different serotypes demonstrate preferred tropism related, in part, to use of varied cellular surface receptors, and cross-species transmission is documented [7]. In the 1980s, AAV was “vectorized” by replacing all viral genes with a transgenic DNA sequence (less than 5 kb) positioned between terminal inverted repeats that serve, minimally, as replications origins. Over the past 3 decades, thousands of reports have demonstrated efficient gene delivery using AAV vectors to a large array of tissue types, which highlights the unique ability of this vector to confer long-term transgene expression in many species [8]. To date, there have been over 1000 human clinical trials based on AAV gene delivery for the treatment of diverse diseases including muscular dystrophies, hemophilia, neurological disorders, and blindness. In gene therapy, AAV particles containing the therapeutic gene of interest enter the target cell, transport to the nucleus, unpackage the cDNA, and form extra-chromosomal episomes, followed by transcription and translation to create the therapeutic protein transgene. Because the cDNA transferred by AAV persists primarily as circular episomes, the transferred gene is not replicated and, therefore, is “diluted” with each cellular division in highly divided cells such as epithelium. Therefore, transducing non-dividing cells, such as corneal keratocytes, ciliary body muscles, and/or cells of the retina are ideal ocular targets to provide long-term gene expression.

Gene transfer using AAV vectors has several advantages particularly when treating ocular disease. The relatively small tissue area, accessible nature of the eye for direct therapy, and overall ocular compartmentalization permit low doses of AAV vectors to produce a therapeutic effect. Furthermore, this low vector dose limits systemic exposure and thus reduces possibilities of an immune response to the viral capsid or transgene. This compartmentalization of the vector and transgene may be particularly important when transferring to the eye transgenes such as HLA-G, a molecule that is expressed by some peripheral neoplasms, allowing them to evade immune surveillance and advance their disease state [9,10]. Currently two AAV therapeutics are FDA approved, including voretigene neparpovec (Luxturna), a gene addition treatment for patients with Leber’s congenital amaurosis [11], a rare form of retinal degeneration, and nusinersen (Spinraza), an AAV gene therapy for spinal muscular atrophy [12]. With the success of Luxturna, there is promise that AAV gene transfer in the eye of an immune-tolerant therapeutic such as HLA-G, may provide long-term effective treatment of chronic ocular immune-mediated diseases following a single-dose administration. In fact, there are 61 ocular clinical trials involving AAV gene therapy currently listed in clinicaltrials.gov.

The purpose of this paper is to review HLA-G presence in the eye, its role in ocular immune tolerance, and recent studies of the therapeutic use of AAV gene transfer in models of ocular disease. Studies reviewed herein include intracorneal AAV-HLA-G administration to inhibit corneal inflammation and vascularization, ex vivo AAV incubation of corneal grafts to prevent corneal transplant rejection, subconjunctival injection of AAV-HLA-G for dry eye and ocular graft vs. host disease, and intravitreal injection of AAV-HLA-G to inhibit uveitis (Table 1). These studies support the theory that AAV-mediated gene transfer of HLA-G may be an effective treatment for common immune-mediated, inflammatory, and neovascular diseases of the eye.

## 2. HLA-G and the Eye

### 2.1. Role of HLA-G in the Eye

There are seven known HLA-G isoforms: G1 to G4 being membrane-associated, and G5 to G7 being soluble. B2m-peptide complexed with HLA-G1 or HLA-G5 isoforms can generate homo-dimers and hetero-dimers that are membrane-bound, free in biological fluids, or present in exosomes [17]. These dimers interact with immune cell receptors Ig-like transcript 2 (ILT2) (CD85j; LILRB1) and ILT4 (CD85d; LILRB2). Other cellular receptors of HLA-G include KIR2DL4, CD8, and CD160 (Figure 1). ILT2 is expressed on monocytes, dendritic cells, B cells, and subsets of natural killer and T cells, whereas ILT4 is expressed by monocytes, macrophages, and some NK cells. HLA-G binding to these receptors results in direct immune cell (NK, neutrophil, T cell, B cell, and dendritic cell) inhibition, reduced proliferation, and prevention of cytokine and chemokine release. Furthermore, HLA-G induces tolerogenic dendritic cells and induction of T regulatory cells that support the ocular immune-tolerant microenvironment (Figure 1) [17].

There is also evidence that HLA-G has an anti-angiogenesis effect in the eye. In an in vivo model of rabbit corneal neovascularization, soluble HLA-G1 inhibited FGF2-induced angiogenesis [18]. We have also demonstrated a significant inhibition of corneal vascularization in a rabbit corneal injury model following AAV-HLA-G1 and HLA-G5 isoform delivery post-trauma (reviewed below) [13]. The mechanism of HLA-G anti-angiogenesis is through direct binding of HLA-G to the BY55/CD160 receptor, which induces endothelial cell apoptosis and/or inhibition of fibroblast growth factor-2-induced capillary-like tubule formation [1].

The immune-suppressive, anti-inflammatory, and anti-angiogenesis roles of HLA-G are consistent with a natural role in ocular immune privilege [13,19,20,21]. Due to its potent activity, HLA-G may have efficacy as a therapeutic in several ocular diseases, such as corneal neovascularization, neovascular glaucoma, uveitis, diabetic retinopathy, age-related macular degeneration, and other diseases of the ocular posterior segment (Figure 2).

#### HLA-G Tissue Locations in the Eye

Because overt ocular inflammation would be detrimental to vision, the eye has developed complex mechanisms to mitigate inflammation and tolerate antigen presence. Similarly to the placenta, where HLA-G functions to allow tolerance instead of rejection of the genetically disparate fetus [22], HLA-G contributes to immune tolerance in the eye [19,21,22,23]. HLA-G expression in tissues comprising the blood–retinal barrier (i.e., RPE), plus the presence of HLA-G protein in the cornea [19], which is the major ocular interface with the environment, all support the role of HLA-G in ocular immune-tolerance (Figure 2).

To demonstrate that HLA-G is present in ocular tissues, several studies have examined the tissue levels of HLA-G in relevant sites of the eye. In one study, total RNA was extracted from human anterior segment tissues (iris, ciliary body, and cornea) and separately, posterior segment tissues (retina, optic nerve, and choroid) and HLA-G expression was investigated using RT-qPCR. HLA-G1, but not G2 or G3, was detected primarily in, collectively, the anterior segment ocular tissues, although it was also detected to a lesser extent in posterior segment tissues [23].

Because the retinal pigmented epithelium (RPE) constitutes the outer blood–retinal barrier, this tissue may express HLA-G as a mechanism to maintain immune privilege similar to that observed in the placenta [22]. Using the human ARPE-19 cell line, it was demonstrated that HLA-G was expressed by normal cultured RPE cells and upregulated when stimulated with proinflammatory cytokines, such as gamma interferon. The authors concluded that HLA-G helps to maintain the immune privilege of the posterior segment of the eye and may be upregulated during inflammation to help restore immune privilege [21].

Immunohistochemistry and RT-PCR for HLA-G were performed on non-diseased human cadaver corneas and diseased corneal tissue removed after full-thickness corneal transplantation, including eyes with keratoconus or pseudophakic bullous keratopathy [19]. Immunohistochemistry in normal and diseased cornea revealed positive HLA-G staining in the corneal epithelium, stroma, and endothelium. RT-PCR showed that HLA-G isoform 1–5 transcripts were detected in the cornea [19].

Collectively, these studies support the hypothesis that HLA-G is present in normal ocular tissues (i.e., corneal, anterior segment, retina, posterior segment tissues), where it helps maintain the immune privilege of the eye.

### 2.2. Development of AAV-HLA-G as a Therapeutic

#### Optimization and In Vitro Validation

A goal in our laboratories was to develop a gene therapy to allow transfer of HLA-G to ocular tissues via an AAV vector [13]. To create this potential therapeutic, initially, HLA-G1 cDNA was codon-optimized for enhanced transgene production in human cells, as well as to remove unwanted alternative open reading frames (ORFs) (Figure 3) [13]. The optHLA-G1 transmembrane domain was then deleted to generate soluble HLA-G1 (termed optHLA-G5 herein) cDNA, and both optHLA-G1 and optHLA-G5 were cloned separately into a self-complementary AAV plasmid context and validated in vitro [13].

### 2.3. Corneal Applications of AAV-HLA-G

#### 2.3.1. Reduction of Corneal Inflammation, Vascularization, and Fibrosis in a Rabbit Model

The ocular surface, including the cornea, is ceaselessly exposed to environmental irritants, injuries, and microbes. To maintain corneal clarity, the eye has developed methods to tolerate the presence of these irritants and microbes, including immunologic tolerance, or privilege, which helps to prevent overwhelming inflammation and subsequent scarring and blindness [2,3,19,24]. Corneal immune privilege is maintained, in part, due to low expression of MHC II molecules, lack of tissue vasculature and lymphatics, few dendritic cells, and presence of immunosuppressive molecules such as HLA-G [2]. Corneal vascularization commonly develops as a result of injury, infection, or genetic disease, and due to its presence, may disrupt immune tolerance, resulting in increased corneal inflammation. In eyes needing corneal transplantation, the compromise of immune privilege (or angiogenic privilege [3]) from trauma, chronic surface inflammation (e.g., dry eye disease), or corneal vascularization results in >70% of the corneal grafts being rejected [25] despite the use of topical and systemic immunosuppressive agents in these “high risk” patients [26]. Although anti-VEGF and other anti-vascular therapies have been attempted to manage corneal neovascularization [27,28], topical corticosteroids are most commonly used in clinical practice [25]. However, use of topical corticosteroids may lead to complications such as cataracts, glaucoma, and toxic keratopathy [29] that can worsen corneal clarity and vision. Thus, there is a high unmet need for the development of improved therapeutics for the treatment of corneal inflammation, vascularization, and prevention of corneal transplant rejection (see Section 2.3.2).

To examine the efficacy of gene transfer of HLA-G to the cornea to prevent corneal vascularization and re-establish corneal immune privilege after injury, we evaluated the intracorneal injection of AAV-HLA-G in the NaOH corneal burn model in rabbits [13]. In this model, central 5 mm corneal burns with NaOH were created, followed 1 week later by central corneal intrastromal injection of scAAV8G9 vectors encoding GFP or a 1:1 ratio of scAAV8G9 HLA-G1 and scAAV8G9 HLA-G5. Animals were followed for 56 days after injury. On both live imaging (GFP) and by immunofluorescence (GFP and HLA-G), wide corneal distribution of the respective transgene was observed. Corneal vascularization was noted in GFP eyes by 7 days after injury and progressed in severity through 56 days. In contrast, eyes dosed with AAV-HLA-G had no significant corneal vascularization develop throughout the study [13]. Histologic and immunofluorescent studies of corneas collected 56 days after injury demonstrated significantly lower inflammatory cell infiltrate, vascularization, and fibrosis in eyes dosed with AAV-HLA-G compared to those treated with AAV-GFP. Furthermore, corneas treated with GFP demonstrated a strong signal in the cornea for alpha smooth muscle actin (αSMA), while corneas treated with AAV-HLA-G were negative. Cells positive for αSMA represent myofibroblasts, which develop in response to injury or inflammation [30]. Myofibroblasts are opaque and produce disordered extracellular matrix in the cornea. When in excess in the cornea, these cells produce fibrosis, scarring, and vision loss [30]. Therefore, we concluded that intracorneal delivery of AAV-HLA-G prevented the differentiation of myofibroblasts and, therefore, reduced development of corneal stromal fibrosis. Collectively, in addition to reducing inflammatory cell infiltrate and vascularization, intrastromal AAV-HLA-G helped to re-establish corneal immune privilege and prevent scarring after injury [13].

#### 2.3.2. Prevention of Corneal Transplant Rejection

Allogeneic corneal transplantation (CT) to treat genetic and acquired corneal blindness is the most common form of tissue transplantation worldwide, annually accounting for >50,000 surgeries in the United States [31]. Corneal rejection, involving a specific immunologic response of the host to the donor tissue, is uncommon compared to other organ transplants due the strong immune privilege of the eye [32]. Nevertheless, this immune privilege can be compromised following CT, leading to rejection rates as high as 20–30% within the first 5 years after surgery [25]. In high-risk situations, which account for >20% of surgeries, there is pre-existing chronic surface inflammation with or without corneal vascularization. In these cases, almost all grafts are rejected within the first 3 years of CT despite the use of topical and systemic immunosuppressive agents to dampen the host’s anti-graft response [25].

Corneal transplant rejection occurs due to cytotoxic T cell-induced apoptosis of the foreign graft, a process assisted by corneal neovascularization. HLA-G is hypothesized to prevent graft rejection by directly inhibiting immune cells involved in foreign tissue rejection and indirectly via vasculature-induced apoptosis and prevention of neovascularization. A practical solution to establish immune tolerance to the donor corneal graft, especially in high-risk cases, is to use ex vivo immune modulatory gene therapy of the donor corneal tissue prior to surgical transplantation [7,14]. With ex vivo donor tissue transduction, only the donor tissue is exposed to the viral vector, which may enhance safety since it obviates the need for vector administration to the donor recipient patient. Using preliminary data in our laboratories, we developed an optimized AAV vector/corneal explant incubation protocol that preserves corneal explant integrity while allowing complete graft transduction [7,14]. Furthermore, we determined that ex vivo HLA-G gene delivery restricts vector genomes to the cornea and does not elicit a neutralizing antibody response to the AAV capsid in primate models [14]. We have preliminary data to support [14] and are in the process of conducting robust studies to determine if ex vivo transduction of donor corneal tissues with AAV-HLA-G isoforms will prevent graft rejection in models of high-risk corneal transplantation.

### 2.4. AAV-HLA-G for Treatment of Ocular Graft vs. Host Disease and Potentially Dry Eye

Over 20,000 patients receive allogenic hematopoietic stem cell transplants (HSCT) per year in the US to treat hematologic disorders [33]. Of these, an estimated 35–54% (approximately 7000–11,000 patients annually) develop ocular graft vs. host disease (OGvHD) following HSCT [34]. The most common clinical manifestation of OGvHD is dry eye disease (DED), which leads to symptoms such as ocular irritation, pain, conjunctival redness, photophobia, and reduced vision [34]. The dry eye of OGvHD is painful and significantly reduces quality of life of HSCT patients [35]. Following the loss of central tolerance post-HSCT, aberrant proliferation of autoreactive T and B cells, autoantibody formation, and a T-cell alloimmune response develop in the lacrimal ductal epithelium, leading to cellular infiltration and inflammation of the lacrimal gland, conjunctiva, and ocular surface [36]. The inflammation can eventually cause a decrease in the density of conjunctival goblet cells, reduce the number of eyelid meibomian glands, as well as induce fibrosis of the lacrimal gland, conjunctiva, and cornea [35,36]. Therapies for OGvHD are directed at reducing symptoms, control of chronic disease, and prevention of tissue damage [34], and include the use of topical lubricants, punctal occlusion, calcineurin inhibitors, corticosteroids, autologous serum, bandage contact lenses, limbal or amnion membrane transplantation, and systemic immune suppressants [34,37]. Despite these treatments, the overall therapeutic response was reported to be only 23% at 6 months [38], indicating a large unmet need for effective treatment of OGvHD.

Murine models of OGvHD have been well-characterized [39] and mimic tear film abnormalities and ocular surface pathology typical of human disease in both OGvHD and chronic immune-mediated DED [40]. An allogeneic murine HSCT model of OGvHD has been established in our laboratories as previously described [41]. Ocular scores (eyelid swelling, corneal opacity, fluorescein retention) and tear production (phenol red test (PRT)) are ocular parameters evaluated following HSCT [41]. One week after HSCT transplantation in this model, AAV8-HLA-G1/5 (1 × 10^9^ vg/eye) was injected subconjunctivally in both eyes. Eyelid and corneal inflammatory and corneal fluorescein stain retention scores in eyes treated with AAV8-HLA-G were significantly reduced compared to a saline control [15]. Additionally, a single dose of AAV8-HLA-G was more effective in controlling signs of OGvHD than twice-daily topical application of cyclosporine (CsA, *p* < 0.02) [15]. These preliminary results indicate that HLA-G has activity in the rodents [16], demonstrating that the subconjunctival therapeutic route targets relevant ocular tissues in OGvHD [42] and provides strong support for the hypothesis that subconjunctival AAV-HLA-G may be an effective single-vector treatment for OGvHD and possibly DED [15].

### 2.5. AAV-HLA-G for Non-Infectious Uveitis

Uveitis is inflammation of the uveal tract of the eye, which includes the iris, ciliary body, and choroid. It is associated with both infectious and non-infectious causes [43]. Non-infectious uveitis (NIU) is a painful disease that is a common cause of blindness in the US, affecting 121 cases per 100,000 adults and 29 per 100,000 children [44]. Normally, the blood–ocular barrier limits the immune response to intraocular antigens. Any disruption of this barrier, either from systemic disease, trauma, or inflammation, can cause this barrier to become leaky, allowing blood products and inflammatory cells to enter the eye [43]. The disrupted barrier also enables various host immune responses to react to self-antigens that are not normally recognized by the immune system. Clinical uveitis often develops spontaneous recurrent bouts of inflammation, likely from T cells recognizing additional autoantigens in the ocular tissue [45]. Recurrence of inflammation leads to progressive destruction and opacification of ocular tissues, resulting in blindness. Treatment for uveitis most commonly is through the use of topical, systemic, or intraocular (i.e., intravitreal) corticosteroids, and systemic immunosuppressant medications, such as cyclosporine, or biologics (e.g., Adalimumab). Although these drug regimens are effective, the side effects can be substantial, including risk of ocular or systemic infection, development of glaucoma, cataracts, ocular scarring, and oncogenesis, among others [5,46]. These therapies are also limited by poor patient compliance and long-term side effects, both of which contribute to the development of blindness [43,47].

One approach to uveitis immunomodulation is to target the auto-pathogenic T cells and inflammatory cytokines that orchestrate and exacerbate the uveitis disease process [48]. There have been several reports of the use of gene transfer for treatment of uveitis, but most of these therapeutics target a single inflammatory cascade or cytokine. Examples include vector delivery of genes encoding IL-10 [49], IL-1ra [50], caspase activation and recruitment domain (CARD) [51], and angiotensin-converting enzyme 2 [52]. Although T-lymphocyte activation and infiltration is the primary mediator of inflammation in uveitis, multiple cell types and inflammatory cascades are involved in the pathogenesis of NIU, including activation of dendritic cells, NK cells, and B lymphocytes in both innate and adaptive immunity [48]. The therapeutic use of AAV-HLA-G may be advantageous over other gene therapy approaches due to HLA-G’s wide and diverse immunomodulatory functions, which can target the multiple immunologic and inflammatory cascades active in uveitis [17]. Additionally, HLA-G’s ability to inhibit vascularization and induce T regulatory function may both help maintain immune tolerance in uveitis and prevent complications associated with inflammatory neovascularization such as choroidal neovascularization, cystoid macular edema, and secondary glaucoma [46]. Thus, AAV-HLA-G would not only treat uveitis via a single injection, but may provide a novel comprehensive therapy for patients to reduce inflammation and prevent complications associated with NIU that lead to blindness.

To evaluate the efficacy and safety of AAV-HLA-G for the treatment of NIU, we examined a well-described and translatable model of NIU, the experimental autoimmune uveitis (EAU) model in rats [53,54]. In these studies [16], naïve Lewis rats received a single intravitreal injection of AAV particles harboring codon-optimized cDNAs encoding the HLA-G1 and HLA-G5 isoforms (1:1 ratio, 2.4 × 10^10^ vg total dose) 1 week prior to the induction of experimental autoimmune uveitis (EAU). AAV-mediated expression of the HLA-G-1 and -5 transgenes in the targeted ocular tissues following a single intravitreal injection of AAV-HLA-G1/5 significantly decreased clinical and histopathological inflammation and cellular infiltration scores compared to untreated EAU eyes [16]. Furthermore, in this model, the anti-inflammatory results after a single AAV-HLA-G intravitreal injection were as effective as daily topical application of dexamethasone. Although effective in controlling uveitis, the corticosteroids had noted side effects, including weight loss and death, that were not observed in the AAV-HLA-G dosed rats [16].

Thus, localized ocular gene delivery of AAV-HLA-G may establish a long-term immunosuppressive effect that would serve as an effective and novel therapeutic strategy for NIU with the potential for applications to additional immune-mediated diseases in the eye.

### 2.6. Limitations and Future Directions

Although there are several reports linking AAV ocular gene therapy to an immune/inflammatory response, especially at high doses, no adverse effects were reported following AAV-HLA-G subconjunctival, intracorneal, or intravitreal dosing [14,15,16,42,55,56,57,58]. Another concern for ocular gene therapy is the durability of the transgene expression following ocular delivery. Factors including viral sequences embedded within the transgenic cassette and the discrete disease etiology may contribute to episomal silencing and/or vector loss following cell death [59]. The durability of vector expression in eye is likely related to the nature of the transduced cells and method of administration, along with the nature of the particular disease. Although HLA-G transgene expression was observed for months in different animal models, additional studies are needed to determine the longevity of HLA-G production in each context [13,15,16]. Regarding potential HLA-G related deleterious effects, in the reported ocular contexts, negative consequences (e.g., increased risk of infection or toxicity) were not observed. Adverse effects of overexpression of HLA-G in the eye are not anticipated because of HLA-G’s natural presence and function in ocular tissues [19,20,21], and, therefore, are likely be subject to natural regulation at least at the protein stability and functional levels. However, long-term studies are needed to infer safety and tolerability in human eyes.

## 3. Conclusions

Work in our laboratories has demonstrated therapeutic efficacy and ocular tolerability of AAV-HLA-G in multiple ocular tissues and in several disease models [13,14,15,16,42,55]. We demonstrated that corneal intrastromal injection of AAV-HLA-G reduced corneal inflammation, vascularization, and fibrosis after corneal injury [7,13]. This corneal application of AAV-HLA-G may also prevent rejection of corneal grafts, especially in high-risk (i.e., vascularized) situations [7,14]. We have also shown that AAV-HLA-G injected subconjunctivally may modulate ocular surface immunity and prevent development of ocular graft vs. host disease and likely DED [15]. Additionally, intravitreal injection of AAV-HLA-G prevented the development of immune-mediated uveitis in a rodent model [16], which provides support for the hypothesis that AAV-HLA-G may be efficacious in human NIU. Together, these studies demonstrate that localized ocular gene delivery of AAV-HLA-G may establish a long-term immunosuppressive effect that would serve as an effective, novel, and safe therapeutic strategy for immune-mediated diseases in the eye.

## 4. Patents

Vector-Mediated Immune Tolerance in the Eye. WO 2018/022683 Al Hirsch ML, Gilger BC.

## Figures and Tables

**Figure 1 ijms-23-03465-f001:**
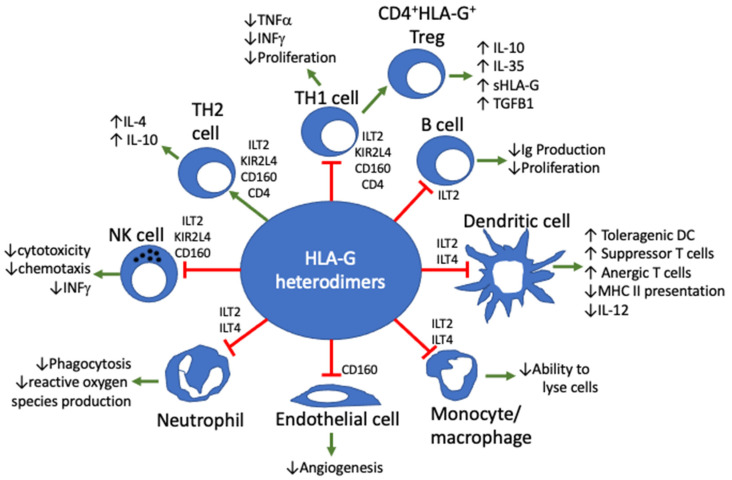
HLA-G actions on various cell types. Cellular receptors and cell activity following HLA-G binding are listed for each cell type. ( ↑ = increased; ↓ = decreased).

**Figure 2 ijms-23-03465-f002:**
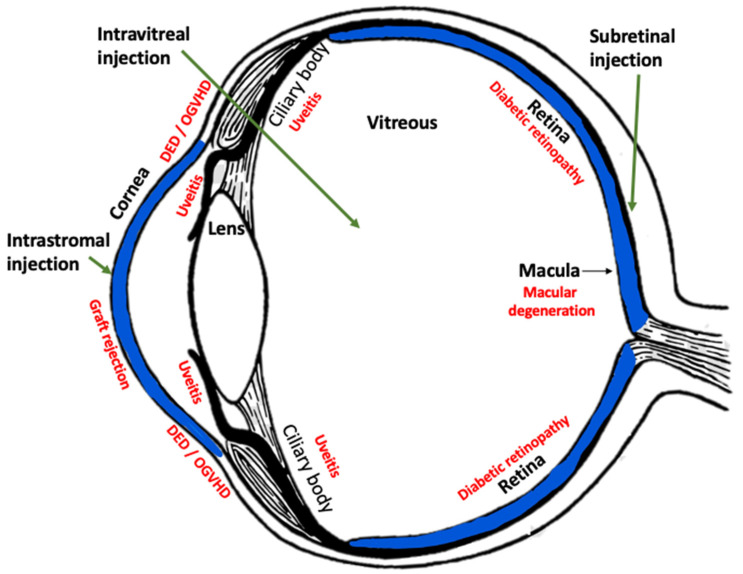
Diagram of human eye demonstrating anatomic structures and sites of injection for AAV gene therapy (green arrows). Blue highlighted areas are where HLA-G has been detected in the normal eye. Red letters indicated common diseases likely to be treated by AAV-HLA-G. DED = dry eye disease; OGVHD = ocular graft vs. host disease (modified from pinpng.com; Accessed on 8 February 2022).

**Figure 3 ijms-23-03465-f003:**
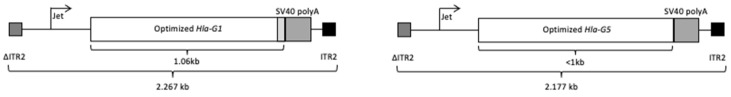
Diagram of the self-complementary scAAV8G9-optHLA-G vector cassette (From Hirsch MH, et al. *Sci Reports* 2017; 7; doi:10.1038/s41598-017-18002-9).

**Table 1 ijms-23-03465-t001:** Publications or abstracts on the use of AAV-HLA-G in the eye.

Target Tissue	Disease	Route of Therapy	Reference
Cornea	Injury/inflammation/neovascularization/fibrosis	Intrastromal injection	Hirsch et al., 2017 [13]
Cornea	Transplant rejection	Ex vivo graft incubation	Bastola et al., 2020 [7] Gilger et al., 2018 [14]
Conjunctiva	Ocular graft vs. host disease/dry eye	Subconjunctival injection	Nilles et al., 2021 [15]
Uvea	Non-infectious uveitis	Intravitreal injection	Crabtree et al., 2019 [16]

## Data Availability

Not applicable.

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
