# Peer review of "Therapeutic Applications of Adeno-Associated Virus (AAV) Gene Transfer of HLA-G in the Eye"

_ijms, 2022, doi:10.3390/ijms23073465_

Round 1

Reviewer 1 Report

Interesting and novel series of studies by the authors on HLA-G function in the eye. Below are some comments:

  1. Authors site own work on use of AAV to induce expression of HLA-G in the ocular tissues and to improve clinical and histopathological outcomes in various ocular conditions. Have they performed flow cytometry to assess HLA-G expression on various cells after AAV injection, or any in vitro assays or in vivo experiments to look at the mechanisms involved through which HLA-G would exert its function in the eye ? such as those mechanisms cited in ref 17 and shown in figure 1. 
  2. Authors' have reiterated own publications and shown figures with details that are already published. This appears bulky and redundant, such as including figures from references 13 and 15 with figure legends. Authors work could have been summarized in short clear sentences within the text only. 
  3. Discussion in the text referring to reference #14 is based on an ARVO abstract published in 2018, is there more data available on the role of HLA-G in corneal transplantation? If so, that's worth mentioning. Also, no mechanisms of action for HLA-G treated donor corneas were reported by the authors; this needs to be mentioned at least as a limitation.
  4. Page 4, line 138, reference #22 does not seem to refer to the eye
  5. Page 9, line 281, reference #38 is 10 years old. Perhaps could add newer reports? such as PMID: 34992343, PMID: 34538201, PMID: 32467449, PMID: 34228599.
  6. Page 14, line 276, among conventional therapies for oGVHD is punctal occlusion (use of punctal plugs or more effectively punctal cauterization) and should be included. 
  7. Were there any side effects observed or anticipated using this treatment approach (AAV) or specifically with increased HLA-G expression in the tissues? Discussion on side effects (whether present or not) need to be added.

Author Response

Thank you for the opportunity to respond to reviewers.  We have provided a point by point response below and incorporated associated changes in the manuscript (noted by tracked changes). 

Please see the responses below (in italic font) and attached revised manuscript.

Response to reviewers:

Reviewer #1

Interesting and novel series of studies by the authors on HLA-G function in the eye. Below are some comments:

Authors site own work on use of AAV to induce expression of HLA-G in the ocular tissues and to improve clinical and histopathological outcomes in various ocular conditions. Have they performed flow cytometry to assess HLA-G expression on various cells after AAV injection, or any in vitro assays or in vivo experiments to look at the mechanisms involved through which HLA-G would exert its function in the eye ? such as those mechanisms cited in ref 17 and shown in figure 1. 

  • As reported in our previous studies, HLA-G expression following AAV vector transduction was confirmed by RT-qPCR and HLA-G protein was confirmed by immunofluorescence of treated ocular tissue. The molecular mechanism(s) of HLA-G function is hypothesized as multi-faceted and has not been discretely studied in different cells of ocular tissues other than the therapeutic effects that were measured by clinical and histological examination in different animal models of various diseases.

Authors' have reiterated own publications and shown figures with details that are already published. This appears bulky and redundant, such as including figures from references 13 and 15 with figure legends. Authors work could have been summarized in short clear sentences within the text only. 

  • We have addressed this concern by removing Figures 4, 5, and 6.

Discussion in the text referring to reference #14 is based on an ARVO abstract published in 2018, is there more data available on the role of HLA-G in corneal transplantation? If so, that's worth mentioning. Also, no mechanisms of action for HLA-G treated donor corneas were reported by the authors; this needs to be mentioned at least as a limitation.

  • The referenced work was discussed in the manuscript as being preliminary findings. Experiments are underway to confirm these preliminary data with rigor and to inform on a potential mechanism(s) of HLA-G.  Minor edits were made to better describe corneal graft rejection and hypothesized mechanisms of HLA-G function in this context.

Page 4, line 138, reference #22 does not seem to refer to the eye

  • This reference refers to a well-studied mechanism of action of HLA-G and is an appropriate citation for the preceding sentence.

Page 9, line 281, reference #38 is 10 years old. Perhaps could add newer reports? such as PMID: 34992343, PMID: 34538201, PMID: 32467449, PMID: 34228599.

  • The 2012 reference was replaced by PMID: 34538201

Page 14, line 276, among conventional therapies for oGVHD is punctal occlusion (use of punctal plugs or more effectively punctal cauterization) and should be included. 

  • Use of punctal occlusion was added to the manuscript as a conventional therapy for OGVHD.

Were there any side effects observed or anticipated using this treatment approach (AAV) or specifically with increased HLA-G expression in the tissues? Discussion on side effects (whether present or not) need to be added.

  • In the reported studies, no side effects were observed following AAV-HLA-G transduction. We added a section on Limitations and Future directions to address this comment and a comment from reviewer 2.

Reviewer 2

AAV based gene therapy definitely has shown great potential addressing ocular genetic disorders. However, as a review paper, I think it will be better to include a limitation and future direction section to discuss the limitations of the AAV based systems.

  • Thank you for this suggestion. We have included a section titled: Limitations and Future Direction – Ocular use of AAV-HLA-G

 As demonstrated in this paper, the example therapies all demonstrated efficacy. However, the evaluations were performed mostly short after gene therapy (less than 56 days). Due to the nature of introduction of a foreign gene, it is quite common for a decay in expression and treatment efficacy, which is observed also for Luxturna. Is this something the author would concern about their AAV therapy? What are the solutions?

  • We have not observed this, but will include a sentence pertaining to this in the aforementioned limitations section

Also, according to the construct map, there are artificial promoter and viral SV40 enhancer sequences in it. Can those sequences cause potential problems for human applications? Is it better to limit the use of artificial or viral gene fractions in the construct?

  • We have not observed a problem in any of our experiments to date, i.e., an immune response. However, we would expect the HLA-G transgene to inhibit an immune response to the vector or construct, thus inhibiting any consequences perhaps induced by viral sequence elements.  However, generally speaking such elements have been associated with inflammation in reports following AAV gene therapy in the eye.

Something minor. Please check the figure quality. I think it might be the system, the figures are blurry. Figure 5 somehow was cropped at the bottom.

  • We will work with the publisher to ensure these are correct. We eliminated figures 4-6.

Reviewer 2 Report

The manuscript “Therapeutic applications of adeno-associated Virus (AAV) gene transfer of HLA-G in the eye” has done a great job in reviewing human leukocyte antigen G (HLA-G) in the eye and its role in immune-tolerance, and the potential therapeutic use of AAV gene transfer and expression of HLA-G in various ocular tissues. Efficacy has been demonstrated from different strategies using different administration routes including intracorneal delivery of AAV-HLA-G to treat corneal inflammation and prevent corneal graft rejection, subconjunctival injection of AAV-HLA-G for ocular graft vs host disease and potentially dry eye disease, and intravitreal injection of AAV-HLA-G to inhibit uveitis. All these together have demonstrated that AAV mediated gene transfer of HLA-G can be promising in treating common immune-mediated, inflammatory, and neovascular diseases of the eye.

The manuscript is well prepared and written. Before moving on to the next step, I have a few questions and hope the author can address.

AAV based gene therapy definitely has shown great potential addressing ocular genetic disorders. However, as a review paper, I think it will be better to include a limitation and future direction section to discuss the limitations of the AAV based systems.

As demonstrated in this paper, the example therapies all demonstrated efficacy. However, the evaluations were performed mostly short after gene therapy (less than 56 days). Due to the nature of introduction of a foreign gene, it is quite common for a decay in expression and treatment efficacy, which is observed also for Luxturna. Is this something the author would concern about their AAV therapy? What are the solutions?

Also, according to the construct map, there are artificial promoter and viral SV40 enhancer sequences in it. Can those sequences cause potential problems for human applications? Is it better to limit the use of artificial or viral gene fractions in the construct?

Something minor. Please check the figure quality. I think it might be the system, the figures are blurry. Figure 5 somehow was cropped at the bottom.

Author Response

Thank you for the opportunity to respond to reviewers.  We have provided a point by point response below and incorporated associated changes in the manuscript (noted by tracked changes). 

Please see the responses below (in italic font) and attached revised manuscript.

Response to reviewers:

Reviewer 2

AAV based gene therapy definitely has shown great potential addressing ocular genetic disorders. However, as a review paper, I think it will be better to include a limitation and future direction section to discuss the limitations of the AAV based systems.

  • Thank you for this suggestion. We have included a section titled: Limitations and Future Direction – Ocular use of AAV-HLA-G

 As demonstrated in this paper, the example therapies all demonstrated efficacy. However, the evaluations were performed mostly short after gene therapy (less than 56 days). Due to the nature of introduction of a foreign gene, it is quite common for a decay in expression and treatment efficacy, which is observed also for Luxturna. Is this something the author would concern about their AAV therapy? What are the solutions?

  • We have not observed this, but will include a sentence pertaining to this in the aforementioned limitations section

Also, according to the construct map, there are artificial promoter and viral SV40 enhancer sequences in it. Can those sequences cause potential problems for human applications? Is it better to limit the use of artificial or viral gene fractions in the construct?

  • We have not observed a problem in any of our experiments to date, i.e., an immune response. However, we would expect the HLA-G transgene to inhibit an immune response to the vector or construct, thus inhibiting any consequences perhaps induced by viral sequence elements.  However, generally speaking such elements have been associated with inflammation in reports following AAV gene therapy in the eye.

Something minor. Please check the figure quality. I think it might be the system, the figures are blurry. Figure 5 somehow was cropped at the bottom.

  • We will work with the publisher to ensure these are correct. We eliminated figures 4-6.

Round 2

Reviewer 1 Report

Authors have appropriately addressed comments and the manuscript appears suitable for publication.